# Physical and financial access challenges to seeking child healthcare in a rural district in Ghana

Felix Kwasi Nyande[1]*, Esmeralda Ricks[2], Margaret Williams[3], Sihaam Jardien-Baboo[2]

1 Department of Nursing, School of Nursing and Midwifery, University of Health and Allied Sciences, Ho, Ghana, 2 Department of Nursing Science, Faculty of Health Sciences, Nelson Mandela University, Port Elizabeth, South Africa, 3 Faculty of Health Sciences, Nelson Mandela University, Port Elizabeth, South Africa

* fnyande@uhas.edu.gh

## Abstract

### Introduction

Timely access to available, and affordable essential child healthcare services is important to ensure the well-being of children under five years. However, healthcare systems in low- and middle-income countries like Ghana struggle to realise this goal because of the accessibility challenges that confront the system.

### Aim

To explore the experiences of nurses and caregivers about the physical and financial challenges to accessing child healthcare services in a rural district in Ghana.

### Methods

The study employed a qualitative approach using an exploratory descriptive design to collect data from nurses and caregivers of children under five years of age in the Nkwanta South Municipality, Ghana. Data collected through semi-structured interviews were analysed using qualitative techniques and the results presented in themes and sub-themes.

### Results

The financial challenges to seeking child healthcare were the high costs of child healthcare services and challenges with health insurance ownership. Limited operational hours, long distances, and transportation difficulties to health facilities were the main health facility and physical access challenges to seeking child healthcare.

### Conclusions

The physical and financial access challenges that confronted caregivers of children under-five years of age contributed to delays and non-utilisation of child healthcare services in various of ways. This situation largely contributed to the inadequate child health services

**Data availability statement:** All relevant data are within the manuscript.

**Funding:** The author(s) received no specific funding for this work.

**Competing interests:** The authors declare no competing interests.

utilisation experienced in rural areas. Addressing these challenges could enhance child healthcare access and improve the state of child health.

## Introduction

Children constitute a vulnerable population at risk of many ailments and adverse health outcomes [1]. Children born Sub-Saharan Africa (SSA) die mostly from preventable causes [2]. Compared to children born in high-income countries, those born in sub-Saharan Africa have a 14-fold increased risk of dying before age five [1]. Inequitable access to child healthcare in Low- and Middle-Income Countries (LMICs), and between rural and urban populations within countries, have been the main drivers of this unfortunate situation ([3,4].

Healthcare systems all over the world seek to protect and promote the well-being of children through the delivery of timely and quality child healthcare services. Timely access to available, and affordable essential child healthcare services facilitates the realisation of this [5]. However, healthcare systems in LMICs like Ghana struggle to realise this goal because of the numerous challenges that they are confront [6–8].

Previous studies have reported accessibility challenges including long distance, inadequate transportation services, and unmotorable roads as physical access barriers to child healthcare services utilisation [9–13]. Several studies conducted in Ghana have reported similar challenges [14–16]. In Ghana where a significant proportion of the population resides in remote and rural areas, the long distances and poor transportation infrastructure make it difficult for families to access healthcare services in time for their children [17]. Additionally, the inability to pay for the cost of child healthcare services put those services beyond the reach of caregivers and their sick children [18–20].

Attainment of the child health target of the Sustainable Development Goal Three in Ghana will be a mirage if the physical and financial access barriers to child healthcare delivery and utilisation are identified and appropriate interventions designed to tackle them. This study aimed to explore the experiences of nurses and caregivers about the physical and financial challenges to accessing child healthcare services in a rural district in Ghana.

## Materials and methods

### Study design, sites, and participants

The study applied a qualitative approach, using an exploratory case descriptive design [21]. The enabled the researchers to gain insight into the experiences of nurses and caregivers about the physical and financial challenges to accessing child healthcare services in a rural setting. Data were collected through face-to-face in-depth interviews.

The setting of the study was the Nkwanta South Municipality, Ghana, a rural district located in the Oti Region of Ghana. Most communities in the district are located several kilometres from the district capital. Approximately 75% of the population in the district live in the rural areas with untarred road networks. Healthcare facilities in the district include two hospitals, several health centres, and Community Health Planning and Services (CHPS) compounds. The hospitals provide both in-patient and out-patient services, while the health centres and CHPS compounds provide only outpatient services. Data were collected in health facilities and communities within the Nkwanta South Municipality.

Nurses who engaged in the provision of child healthcare services and caregivers of children under five years of age constituted the study population. The caregivers were either users or non-users of the available child healthcare services. Participants were purposively sampled

from the two hospitals, a health centre and three CHPS compounds. These facilities were purposively selected to include the different levels of service delivery at the district level in Ghanaian health system, that is levels A, B and C. With the assistance of the Municipal Health Directorate, eight communities were also purposively selected and included in the study because these communities had high number of caregivers who did not regularly utilise the available child healthcare services. Based on the eligibility criteria, nurses and caregivers at health facilities were identified and included in the study. Selection of participants from the communities was done with the assistance of community health volunteers in the primary healthcare facilities based on the eligibility criteria. The sample size of 26 participants was made up of ten nurses, nine caregivers who utilised the available child healthcare services, and seven non-user caregivers.

Recruitment of participants and data collection spanned January 9th, 2019, to March 22, 2019. The participants were purposively selected based on their experiences with child health-care services in the study area. Nurses who worked in health facilities within the municipality and had been directly involved in the provision of child healthcare services for at least six months were eligible for inclusion in the study. For the caregivers who utilised the available child healthcare services, the eligibility criteria were: caregiver of a child less than five years of age who resided in the municipality and attended any of the public health facilities to access child healthcare services. Caregivers who had not used the available child healthcare services within the past year even though there was a need to do so were included as non-users.

## Data collection

The field investigator (FKN), a PhD nursing student at the time, collected the data. Semi-structured interview guides (one for each group) were used to collect data from the participants. A separate form was used to collect baseline demographic data of each participant. At the health facilities, the field investigator was introduced by the nurse managers to the nurses and caregivers accessing child healthcare services before data collection. Community Health Volunteers in the health centres and CHPS compounds and assembly members of the electoral areas introduced the field investigator to caregivers in the communities. The field investigator extended an invitation to the participants to take part in the study. A suitable time and place for the interviews were agreed upon by the field investigator and the participants before the interviews. In the health facilities, interviews with participants were conducted in open areas under trees situated within the premises, where there were no interferences from other people. A few interviews were conducted in the offices spaces where no one interfered. This approach was adopted to minimise interruptions and to ensure the privacy of the participants. In community settings, caregivers were interviewed in either a secluded corner or outside the main compound, based on which environment afforded a more conducive atmosphere for open and free expression. This careful consideration of the interview settings aimed to facilitate candid dialogue while protecting the confidentiality of the respondents.

The purpose of the study was explained to each participant before the commencement of the interviews. Sixteen interviews comprising ten nurses, and six of the caregivers who utilised the available child healthcare services were conducted in health facilities, with the remaining nine interviews with the caregivers being conducted in the communities. The sample size was controlled by data saturation; the point at which the addition of participants produced no new data [22]. Because data collection and analysis were done simultaneously, at the point of data saturation, redundancy of codes were realised, as no new codes emerged from the additional data collected.

Interviews conducted with the nurses and four of the caregivers were in the English lan-guage, and the rest were conducted in the *Twi* language, a local Ghanaian language spoken in

the study area. The interviews were recorded using a digital audio recording device and were played back to the participants for clarification where and when necessary. The audio recordings were transferred to the personal laptop of the field investigator at the end of each day for storage and processing. Field notes and reflections, made during the data collection were incorporated into the interview transcripts before data analysis commenced.

## Data management, control and analysis

After each interview, the audio recordings were transferred to the personal laptop of the field investigator and secured with a password. The files were sorted according to the participant groups and locations; separate files were created for each population of the study. The researchers listened to the audio recordings of the interviews several times to interact with the raw data. This ensured familiarisation with data before coding started and also enabled the researcher to seek clarification from participants were ambiguities arose. The audio recordings of interviews conducted in English were transcribed verbatim. Those interviews conducted in Twi were transcribed first and subsequently translated into the English language.

The coding and organisation of the data were done by the field investigator and an independent coder. The independent coder is a seasoned qualitative researcher with a PhD in nursing. Data analysis was done using content analysis. The steps of coding described by Tesch [23] were applied. Following the initial examination of the raw data, each interview transcript underwent multiple careful readings. Open and in vivo codes were derived from a list of identified topics. Subsequently, these codes were synthesised into relevant axial codes using precise terminology. Clusters of analogous topics were organised into columns, facilitating the identification of sub-categories, sub-themes and themes, which were then aligned with their respective descriptive topics (table 1).

## Ethical considerations

Two research ethics committees approved the study protocol before data collection: the Research Ethics Committee of the Nelson Mandela University (reference number: H18-HEA-NUR-018), and the Ethics Review Committee of Ghana Health Service (reference: number GHS-ERC014/11/18). Administrative approvals were granted by the Volta Regional Health Directorate, the Nkwanta South Municipal Health Directorate, and the Nkwanta South Municipal Assembly before data collection. Permission was obtained from the management of each of the hospitals involved in the study.

Written informed consent was obtained from each participant, before participation. The contents of the participant information sheet were verbally translated into Twi for those participants who could not comprehend the English language; this was done in the presence of a witness, who also signed the consent form to indicate that the translation and all clarifications were done accurately. To uphold participants' right to privacy and confidentiality, participants' responses were made anonymous.

## Trustworthiness

The rigor of the study was ensured by applying the criteria of credibility, dependability, transferability, and confirmability as outlined by Lincoln and Guba (cited in Polit & Beck, 2017) [24]. To ensure credibility, the field investigator went strictly by the interview guide during data collection. As part of reflexivity, he stuck to the interview's topic. By collecting data from different stakeholders in child healthcare, we captured the differing perspectives on the research question. An independent coder's engagement to code the data further enhanced the study's credibility. The entire research process has also been vividly described to facilitate

**Table 1. Codes, sub-categories, sub-themes and themes from data analysis.**

| Codes | Sub-themes | Themes |
|---|---|---|
| - Lack of regular income source<br>- No support social system<br>- Feeding cost<br>- Cost of transportation<br>- No personal means of transport<br>- Desire to use services not backed by means of affordability<br>- Anticipated cost of referrals scare caregivers<br>- Lack of readily available market for caregivers' farm produce<br>- Caregivers' source of income tied to seasonal nature of their occupation | - Affordability challenges<br>- Refusal of referrals due to financial challenges | Financial access challenges |
| - No money for health insurance registration<br>- Inability to afford out of pocket payment<br>- Late reporting to health facility leading to poor child health outcomes<br>- No money for health insurance renewal<br>- Low importance attached to health insurance renewal<br>- No NHIS offices within the communities<br>- Poor internet challenges during registration causing delays<br>- Caregivers make additional charges even with NHIS<br>- Caregivers feel health facilities exploit them even when they have NHIS<br>- Caregivers feel the quality of care is reduced with NHIS | - No Health Insurance<br>- Expired NHIS membership<br>- Dissatisfaction with NHIS coverage | Health insurance challenges |
| - Health facilities located too far from communities<br>- Caregivers find it difficult to reach facilities on foot and at night<br>- No personal means of transport<br>- No means of transport during emergencies<br>- High cost of transportation<br>- Use of motorcycles and bicycles<br>- Bad road network during rainy season<br>- Flooded road networks<br>- Vehicles move only on specific days/market days<br>- Delays in waiting for vehicles | - Long distances to health facilities<br>- Unreliable public transport and bad road network | Physical access barriers |
| - Clinics operate only during the day<br>- Services not available after closing hours<br>- Refusal to attend to patients outside operational hours<br>- Resorting to self-medication and traditional medicine use | - Limited clinic operational hours | Health facility barrier |

replication of the study in similar settings or with similar participants, to uphold the criterion of transferability. The inclusion of direct quotes from participants in the data analysis is to uphold confirmability. The independent coder and audit trail ensured the dependability of the study.

# Results

## Participants' characteristics

The nurse participants were within the age range of 27–35 years. Six out of the ten nurse participants were females. These participants came from diverse professional backgrounds within nursing; three were Registered General Nurses, two each were Registered Nurse Assistants (Clinical), Registered Nurse Assistant (Preventive), and Registered Community Health Nurses respectively, and one Registered Midwife. Two of the participants held an additional qualification in paediatric nursing.

All the caregivers were female, and all took care of their children except one who cared for her grandchildren. Five caregivers attended basic education; four completed senior high school, only one participant had a tertiary education qualification; the remaining six had no formal education. Two of the caregivers were unemployed while ten of them were peasant farmers; with the remaining four being a teacher, a baker, a hairdresser, and a healthcare

volunteer. The majority of them (10) were Christians, two were Muslims, and the remaining four were Traditionalists.

### Themes and sub-themes

Four themes and eight sub-themes emerged from the data analysis as depicted in table 2 below.

### Theme one: Financial access challenges

It emerged from the qualitative interviews that financial challenges hampered the utilisation of child healthcare services by caregivers of children under-five years of age. The financial access challenges experienced by the caregivers centred around the cost of healthcare services; the lack of health insurance; and health insurance challenges for those who are insured. These challenges were experienced by both caregivers who utilised the available child healthcare services as well as those who did not, even when there was the need to.

### Affordability challenges

Accessing healthcare services comes with costs directly associated with payment for the healthcare services. Caregiver participants described the cost of child healthcare as the reason they could not utilise the available child healthcare services. The non-user caregiver participants particularly explained that financial difficulties constituted the primary reason for their non-utilisation. A caregiver who did not utilise the available child healthcare had this to say:

> *The first [reason for non-utilisation] is money. If I have the money, there is no problem sending my child to the hospital. But since I don't have money, I can't [take the child to the hospital] (Non-user 2).*

The caregivers' assertion was confirmed by nurse participants who mentioned that some caregivers failed to utilise child healthcare services due to financial constraints. This quote from one of the nurses at the hospital illustrates this point:

> *…it is money. They (caregivers) don't have money to come [to the hospital] (Hosp. nurse 2).*

The caregivers expressed their desire to utilise available child healthcare services when needed, however, their inability to afford the services made it impossible. The unaffordability of services ranked highest among reasons why caregivers failed to utilise child healthcare services. The following quotes from some of the caregivers buttress this point:

**Table 2. Themes and sub-themes on physical and financial challenges to accessing child healthcare services in a rural district in Ghana.**

| Themes | Sub-themes |
|---|---|
| 1.  Financial access challenges | 1.1  Affordability challenges<br>1.2  Refusal of referrals due to financial challenges |
| 2.  Health insurance challenges | 2.1  No Health Insurance<br>2.2  Expired NHIS membership<br>2.3  Dissatisfaction with NHIS coverage |
| 3.  Physical access barriers | 3.1  Long distances to health facilities<br>3.2  Unreliable public transport and bad road network |
| 4.  Health facility barrier | 4.1  Limited clinic operational hours |

*The money issue is the whole problem. If we are able to raise some money, we will take her (sick daughter) to the hospital. Hmmmm... (shakes her head; chokes on her words...) hmmmm... hmmmm... hmmmm... it all boils down to money. It is money that is the problem (Non-user 7).*

*Because if I take him to the hospital, they will ask me to make payment and I don't have any money on me to make any payment (Non-user 1).*

The anticipated cost of child healthcare services made some of them stay away from utilising the services.

*I think that when I go [to the hospital], money matters will arise. When you send her and without money, the doctor cannot attend to the child, there's no need sending her. This is because I don't have money to pay the doctor (health facility) (Non-user 3).*

The nurse participants described situations in which they encountered caregivers who kept their sick children at home because they could not afford the cost of child healthcare services. The nurse participants further explained that, without health insurance, patients who visited health facilities are required to make a cash deposit before they are attended to, however, these caregivers could not afford the cost of this deposit. A nurse at one of the primary healthcare facilities had this to say:

*Sometimes too they [caregivers] tell you that there was no money [to bring the sick child to the facility] (PHC nurse 3).*

The indirect and opportunity costs of accessing child healthcare services also hindered the smooth prompt utilisation of child healthcare services by caregivers. Participants described ancillary costs such as transportation charges, feeding, and associated costs which added to the total cost of accessing child healthcare services. Some participants explained the situation in the following quotes:

*… you would need to board a car or motorbike, so you have to look at how much [money] you have. Maybe after the doctor has attended to you, they may tell you some things that need to be done for your child and will have to be paid for (Caregiver 3).*

*One of their (caregivers) reasons [for the delay in coming to the hospital] is money, and since we have a lot of villages around, boarding a car from the village to this place, they don't have the money to do so (Hosp. nurse 1).*

Some participants explained that these extra costs, particularly for transport, accounted for the delayed or non-utilisation of child healthcare services. A nurse had this to say:

*Mostly they use motorcycles and most of the families are not having these motors in their homes, so mostly they have to pay, and they do not have the amount of money to bring the children to the hospital (Hosp. nurse 1).*

## Refusal of referrals due to financial challenges

It became apparent from the interview that the referrals to higher-level facilities came with additional financial demands for the families of sick children. The cost of referral thus, caused caregivers not to honour referrals of their sick children to higher facilities for appropriate treatment.

Participants stated emphatically that financial challenges were the main reason why caregivers failed to honour the referral of their children to higher-level healthcare facilities. A nurse participant highlighted the situation in the following quote:

*Sometimes when we refer them [to the next facility], they do not go. When they come again, and we ask them why they did not go, they will tell us they didn't have money to go (PHC nurse 2).*

The caregivers usually return home with the sick children instead of continuing to the higher-level facility that they have been referred to. A caregiver whose daughter was referred to the regional hospital narrated how the lack of money caused them to take the sick child home instead of to the referral facility. They anticipated the monetary demands that they would encounter hence the decision to take the sick child back home instead. The lamentations of one of the caregivers is captured in the following quote:

*Hmmmm! There is no money. You see when you are travelling, you need money to travel. If we don't have any money and we decide to take her to the Regional Hospital, maybe we will get there, and several issues will come up that demand money "(Non-user 7).*

Farming is the main occupation of families in the study area. They thus, depend on the sale of their farm produce to meet their economic needs. However, the lack of a readily available market, coupled with the seasonal nature of farm produce, hampered their ability to pay for child healthcare services. Until the farm produce is harvested and sold, they are unable to raise sufficient funds to cover the costs of child healthcare. Nurse participants who regularly interacted with the community members made similar observations regarding the role of the occupation of caregivers in their inability to child healthcare services. A nurse participant explained the cause of this state of affairs as captured in this quote:

*Well, most of the parents are farmers and their crops are annual crops so unless the season comes, they don't have money. So, if the money is not there maybe if their crops are not yet sold, they don't have, and the children happen to fall sick, since they don't have money, they keep the children in the house (Hosp. nurse 1).*

## Theme two: health insurance challenges

The National Health Insurance Scheme (NHIS) is a social insurance scheme that grants financial access to its subscribers. Children under eighteen years of age are covered by the registration of their parents and hence enjoy access to healthcare services under the scheme. However, without active NHIS membership, parents and their children are supposed to bear the full cost of healthcare services when they visit health facilities. NHIS challenges described included not possessing an active NHIS membership, challenges experienced in enrolling and renewing one's membership as well being denied benefit packages at the point of usage.

## No health insurance

The participants described the lack of active NHIS membership either as a result of non-registration as causing caregivers to fail to use child healthcare services. This situation was confirmed by non-user caregivers. Below is a quote from some of the participants, to lend credence to this point:

*It's because I don't have health insurance. If I send him (my son) to the hospital they will take money from me. And my husband does not also have… if you do not have health insurance, when you go to the hospital, you may not be attended to because you cannot pay (Non-user 3).*

Nurses alluded to the fact that some caregivers reported to health facilities very late because of a lack of health insurance. These caregivers usually resorted to other alternative treatments and only turned to health facilities when they realised that the child's condition was not improving. Caregivers only rushed their sick children to healthcare facilities after the child's condition had deteriorated. This quote throws light on the situation:

*Some (caregivers) too don't have health insurance, so when the child is sick, they will put the child down at the prayer camp or they will resort to herbal preparations. If the child does not improve, and they see the signs that the child will die, then they quickly take a motorbike and rush the child here for treatment (Hosp. nurse 2).*

### Expired NHIS membership

Annual renewal of NHIS is required for subscribers to continue to enjoy the benefits. Some caregiver participants explained their subscriptions had expired rendering them unable to access child healthcare. This statement by a nurse in hospital highlights this situation

*At times, they* (caregivers) *will say health insurance and financial problems. They have not registered for health insurance, or they have registered, and it has expired (Hosp. nurse 3).*

Financial difficulties encountered by some caregivers made it difficult for them to renew their NHIS membership. Renewal membership requires that NHIS subscribers pay a renewal fee to activate their expired membership. A caregiver had this to say:

*When I was pregnant I did [NHIS]. You know that one is free. So subsequently I did not get money to do it for the children… I wish I could renew my NHIS subscription). It's the financial constraints that is the problem (Non-user 3)*

On the contrary, some nurse participants viewed the failure of caregivers to renew their NHIS subscription to be because of the low importance they attached to it.

*I do not know if the poverty affects the NHIS subscription too but, because they don't really value it unless they are sick. When it comes to renewal, they don't normally do (Hosp. nurse 5).*

Both nurse and caregiver participants further blamed the unavailability of NHIS registration offices in the communities for some families' inability to renew their NHIS membership. Also, specific challenges including internet connectivity (network) challenges made it difficult for caregivers to subscribe or renew their subscription to the NHIS. Some participants are quoted below to highlight the situation:

*We would have done it (NHIS registration) long ago, but they don't do some in this town, unless at Nkwanta. [It is far] yes, because it all involves money (Non-user 5).*

*The NHIS system here too is not the best; sometimes the network is down (Hosp. nurse 5).*

### Dissatisfaction with NHIS coverage

Caregiver participants lamented the limited coverage of child healthcare services by the NHIS. Thus, they were compelled to pay for services that were not covered under the NHIS. This meant extra cost to the caregivers even with NHIS membership. One caregiver's explanation of the situation is captured in the quote below:

> *Even though my son has health insurance, it does not cover everything. When you come* [to the health facility]*, before they will diagnose your child, you have to pay some charges before they start (Caregiver 2).*

Caregiver participants lamented the limited coverage of child healthcare services by the NHIS. According to them the decision to take their sick child to the health facility hinged on their being able to gather enough more considering the extra cost they could incur even with an active NHIS membership. Thus, they sometimes either delayed or failed to utilise child healthcare services if they had no money for the extra charges.

> *Even though there is* [health] *insurance, you get there (hospital) sometimes, and some drugs will be prescribed for you to go and buy. I may not have money at that moment to buy drugs or there is nobody to ask for money from or assist me buy those drugs. … but I may not have money (Caregiver 8).*

Some caregiver participants felt swindled and accused healthcare workers of deception regarding the scheme's coverage. They also perceived that the NHIS deliberately extorted monies from them for services that are supposedly covered under the scheme. To these caregivers, the NHIS service coverage was not spelt and hence haphazardly implemented in the healthcare facilities:

> *Hmmmm, for the health insurance, they will tell you that you won't pay anything but that is not true. You see, maybe the bed that you will lie on at the health facility that you may not pay for. They may write a prescription for you to go and buy medication and these medications may be very expensive (Non-user 7).*

### Theme three: Physical access barriers

The participants described the limited accessibility of services as hindrances to the utilisation of child healthcare services in the Nkwanta South Municipality. They described challenges including transportation and health facilities that are located too far from the communities.

### Long distances to health facilities

The distance to the nearest health facility emerged as limiting prompt and consistent utilisation of child healthcare services. According to the caregiver participants, the available healthcare facilities are located too far from their communities. Thus, making the journey to the nearest healthcare facility on foot could sometimes take up to two hours, which was a cause of concern for them. Caregiver participants who had no means of transport journeyed on foot carrying their sick child whenever they had to utilise child healthcare service. The following quotes from some of the participants lend credence to this situation:

> *The other problem we face is transportation… we have to walk from here to NKwanta because there is no vehicle (Caregiver 4).*

*Please, I walk from the village to town (nearest health facility). I carry my child and walk the full distance from the village to town. Sometimes it is* [a] *worry to me, but I have no choice (Caregiver 6).*

The caregiver participants narrated that they had to endure long distances on foot to get to the nearest healthcare facility. According to them, the situation was worse at night when public transport hardly operated. Some caregivers are quoted below:

*If your child falls sick at night, there is no motorbike, car, or anybody to help you get to the clinic. Walking from here to the clinic too is far (Non-user 6).*

*If you are walking, it can take you about two and half hours; with your sick child. So if you want to walk with your child, then you have to set off very early in the morning; like 5am then you start your journey, by 7 o'clock or 7:30am you will be at the hospital (Caregiver 2).*

Some of the communities are located far from the health facilities so caregivers are unable to make the journey on foot. In these communities a means of transport is required to reach the health facility. The caregiver participants expressed worry because most of them do not have their means of transport, nor do they have the money to pay for the services of commercially available transport.

*If you don't have the means [of transport], then you have to go and look for a source or means of coming to the hospital, that is why I said the transport and transportation are issues of concern (Caregiver 2).*

*If you don't have the money to pay for a car or motorbike to take you to Nkwanta, then it becomes a problem. You see, I can't carry the two of them (twins) and walk from here to Nkwanta in the sun. The distance to Nkwanta is not that close (Non-user 7).*

The participants unanimously agreed that the long distances to healthcare facilities caused caregivers to either delay in visiting healthcare facilities or resort to home remedies. The following quotes from some of the participants explain the situation:

*I think genuinely some of them (caregivers) will like to bring their children to the hospital, but they feel like the place is very far, and travelling all this distance they would want to finish organising themselves before they travel [leading to delays] (Hosp. nurse 5).*

*Means of transportation and the distance from the nearby villages to the health centre [are obstacles]. They mostly use motorbikes. The women don't ride the motorbikes; it's the men who ride and bring them. So, if the man is not at home, they wait until the man comes and says that he's carrying them to the hospital, before they bring the child to the hospital (Nurse 10).*

## Unreliable public transport and bad roads

The absence of a readily available public transport system was yet another obstacle that hampered the utilisation of child healthcare services. Caregiver participants explained that they endured long waiting hours to get transported to the healthcare facilities, irrespective of the severity of the child's illness. According to caregiver participants, vehicles travelled to some of the communities only on certain days and times of the week. They explained the situation in these quotes:

*In the villages, it is not easy; even if you have the money, it is not that easy to get the transport… For instance, in my area, it is only on Mondays, and Sundays, which is Keri and Nkwanta market days, respectively. Apart from these days, vehicles do not move to and from the village (Caregiver 2).*

*When you get there (station) and it is not market day, the car will not get full early, so you have to wait until it is full because it won't pick you alone (Caregiver 4).*

The bad nature of the road network in the area contributed to an erratic public transport system. Getting to the healthcare facility was a challenge for caregivers when the sickness of a child did not coincide with the movement of vehicles to and from their communities; thus, leading to delays in accessing child healthcare services.

*Getting a vehicle to Nkwanta [municipal capital], at times you have to wait for a very long time before and the road too is bad (Caregiver 7).*

*The roads, [for] most of them (caregivers) where they come from, the roads are not accessible. They only get means of transportation during market days (Hosp. nurse 4).*

The road network connecting the communities becomes unmotorable during the rainy season, as streams and rivers flood the already bad roads. Some of the roads became so bad that it was nearly impossible for cars to drive on them, except for motorbikes or bicycles. Caregiver participants expressed fear and safety concerns when they ply these bad roads while carrying their sick children. A description of the situation is captured in the quotes below from two of the caregivers:

*The roads are not safe, but we manage. Because from Nkwanta to our village is very dangerous. Sometimes if you are on a motorbike, you will get to a place where you have to get down and walk before you continue on your motorbike again. The road is not safe at all. There are some lakes or rivers on the road. When it is rainy season they flood and cross the road, so you have to walk (Caregiver 2).*

*You see, a motorbike or a tricycle cannot use this route; it is only a bicycle that can… If you take a motorbike for instance, the road is so bad that by the time you get there, you won't like it (Caregiver 6).*

However, the use of these other means was not suitable for the transporting of patients. It is more difficult for caregivers to access child healthcare services during the rainy season, because of the bad road network that hampered transportation. Some caregivers threw on light in the following quotes:

*Hmmm, there are some big streams to cross in the rainy season, and that is not a joke. It is difficult to take the sick child to the health facility during the rainy season (Caregiver 6).*

*My husband picked us on his motorbike and the road too wasn't good, so we wasted time. It took us about an hour. Because the child is not feeling well, he couldn't ride with speed. The road wasn't that good so he [husband] rode us patiently till we got to the hospital (Caregiver 4).*

According to some caregiver participants, it was practically impossible for one person to transport a sick child on a motorbike to the health facility on the bad road network. It required at least two people to transport a sick child to the health facility; the pillion observes the sick child with the other riding the motorcycle. A caregiver explained the situation in this quote:

*When you take a motorbike you get someone who will sit behind so that he or she can support you with the patient. At times you will manage her [patient] by carrying her on your back. That is what I said earlier, someone will be seated at your back looking at how her condition is until you get there (Caregiver 5).*

### Theme four: Health facility barrier

The participants described the limited availability services as a hindrance to the utilisation of child healthcare services in the Nkwanta South Municipality.

### Limited operational hours

It emerged that the PHC clinics located in the communities operated only during the day, limiting the availability of services to only those hours of the day. According to caregiver participants, they could not access child healthcare services outside these operational hours, especially at night when the clinics are closed. The caregiver participants alluded that healthcare workers turned away patients during non-operational hours. One caregiver who is not regular user of the available child healthcare services explained her reasons in the quote below:

*Usually, when it is 6 pm, then the hospital is closed. There will not be any nurses or doctors that will attend to patients… For instance, last time, one of my children was sick at dawn, and I together with my in-law, sent her* (sick daughter) *to the hospital and we knocked at the hospital's gate for a very long time, and yet no one came to our aid... We went back* [to the health facility] *around 7am to 8am, and yet there was no nurse or doctor to attend to us (Non-user 5).*

This situation forced caregivers to resort to alternative means of treatment for their sick children. These other options included over-the-counter self-medication. The caregivers found these alternative options more convenient because they were always available and ready to attend to patients irrespective of the time of day. Some other expressed their frustrations as captured in the following quotes:

*…meanwhile, the person is also suffering. So, I will take the person to a place that they can take care of him/her even if it is early in the morning (Non-user 4).*

*When I go to the facility and I don't meet the nurses, I send them* (sick children) *to the drug store and inform the druggist about the condition (Non-user 3).*

Non-user caregiver participants described that they did not even attempt to visit the healthcare facilities during certain times of the day when they knew that the facilities were closed.

## Discussion

We explored the experiences of nurses and caregivers of children under five years of age about the financial and physical access challenges to accessing child healthcare services in a rural district in Ghana. We found the financial challenges that impeded access to child healthcare to include the direct and indirect cost of healthcare services, health insurance ownership, and related challenges. The costs associated with child healthcare services were beyond the capability of many caregivers hence their inability to access the services when needed. Also, the ancillary costs to child healthcare service utilisation such as transportation and feeding costs, which caregivers incur when they try to access these services outside of their communities hindered the timely utilisation of child healthcare services in the study area. Our findings

are consistent with previous studies which found the cost of healthcare services to be the main reason accounting for the non-utilisation of child healthcare services especially among the poor [25–29]. Express desire to utilise child healthcare services must be backed by the ability to pay for those services. Despite the desire expressed by care to utilise child healthcare services when needed, the financial challenges they faced limited the prompt utilisation. Invariably, the delays or non-utilisation of child healthcare services among rural dwellers due to financial barriers contributed to the poor state of child healthcare experienced in these areas.

The study further found that some caregivers turned down referrals of their sick children to higher-level health facilities because of the uncertainty of the cost involved. Singh et al. [30], reported similar findings that caregivers in Ghana refused referrals because they were uncertain of the cost implications of these referrals and could not raise adequate funds for said referrals. Caregivers' failure to utilise child healthcare services including referrals because they cannot afford the cost, could contribute to poor child health outcomes.

It further emerged that caregivers not possessing active NHIS ownership impeded smooth access to child healthcare services. Health insurance ownership improves financial access to healthcare services by reducing catastrophic expenditure [31]. The cost of child healthcare is reportedly higher for uninsured households, as they understandably make more out-of-pocket payments at the point of service [32,33]. Additionally, the challenges experienced by caregivers in the registration and renewal of the NHIS membership sometimes prevented eligible members from enrolling in the scheme. These challenges included distance to the nearest NHIS office and the occasional technology failure leading to the temporal unavailability of NHIS services. These findings are consistent with previous studies [18,34]. Again, caregivers felt NHIS offered limited range coverage of child health services. Caregivers therefore bear the costs of those services not covered by the scheme, thus increasing the cost of child healthcare services. Caregivers seemed unsure of which services fell within the exclusion or inclusion criteria, hence their disagreements with healthcare workers. These findings are consistent with those of previous studies in Sub-Saharan Africa [29,35,36]. Even though NHIS membership may significantly reduce the financial burden on households and families, the poor could still be at risk of catastrophic health expenditures owing to limited coverage of NHIS services.

The caregivers also experienced challenges involving health facility and physical access to healthcare services which impeded the utilisation of these services in the Nkwanta South Municipality. Primary healthcare clinics located close to caregivers operated limited hours which reduced the availability of their services. These findings are consistent with those reported by previous studies [16,29,37,38]. The availability of health facilities could influence care-seeking behaviour by caregivers of sick children. Hence, the unavailability of healthcare services especially at night discouraged caregivers from utilising these services. It was found that most of the healthcare facilities are located too far from the communities where caregivers and their children reside. Caregivers therefore had to travel long distances to get to these facilities. Previous studies conducted in rural Ghana and elsewhere discovered that long distances made it difficult for caregivers to journey to healthcare facilities at the onset of their child's illness, especially at night [37,39–41].

It further emerged that persistently unreliable public transport system delayed access to child healthcare services. Caregivers either waited for long hours to get public transport to health facilities or made the journey on foot. These findings support previous studies that have also found that the lack of transport hampered access to child healthcare services, especially among rural dwellers [15,29,42]. The transportation challenges experienced by the caregivers are largely due to the bad road network in this area. The Nkwanta South Municipality is a predominantly rural area, with untarred road networks; hence most communities in the district are cut off from other communities during the rainy seasons causing difficulty in vehicular

mobility [43]. Previous studies have also found that poor road networks and a lack of transportation hindered access to child healthcare services in rural areas [40,42,44,45].

## Conclusions

The financial and geographical access challenges to child healthcare services utilisation uncovered in this study are reflective of the socio-economic and infrastructural challenges that confront typical rural dwellers. The findings point to the devastating impact of these barriers on child health particularly, among rural and marginalised populations. These barriers significantly contribute to and continue to sustain the widening gap in child healthcare outcomes between urban and rural communities by contributing considerably to the delay and non-utilisation of child healthcare services. Also, caregivers resorting to alternative treatments including self-medication to overcome these challenges could have consequences on child health outcomes. National efforts aimed at achieving UHC could be derailed by these challenges.

To improve child health outcomes and achieve UHC, these challenges need to be promptly addressed. Concerted efforts targeted at caregivers and their families to encourage enrolment onto the NHIS will help reduce the financial barriers to child healthcare services. Also, there is a need to address the challenges caregivers face regarding NHIS enrolment and membership renewal to improve their experiences and make the scheme more effective in responding to the needs of subscribers. These strategies will help reduce catastrophic health expenditure caregivers experience because of out-of-pocket payment for healthcare services.

Again, there is the need to increase the number of healthcare workers at the PHC clinic so that they can render child healthcare services around the time. Furthermore, an improved road network is necessary to address the geographical access barriers, reduce waiting and travel time to the health facilities as well as enhance the early and prompt utilisation of child healthcare services.

## Author contributions

**Conceptualization:** Felix Kwasi Nyande, Esmeralda Ricks, Margaret Williams, Sihaam Jardien-Baboo.

**Data curation:** Felix Kwasi Nyande.

**Formal analysis:** Felix Kwasi Nyande.

**Investigation:** Felix Kwasi Nyande.

**Methodology:** Felix Kwasi Nyande, Esmeralda Ricks, Margaret Williams, Sihaam Jardien-Baboo.

**Supervision:** Esmeralda Ricks, Margaret Williams, Sihaam Jardien-Baboo.

**Writing – original draft:** Felix Kwasi Nyande.

**Writing – review & editing:** Felix Kwasi Nyande, Esmeralda Ricks, Margaret Williams, Sihaam Jardien-Baboo.

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
