## [Decision Letter · Decision Letter 0]

10 Jul 2024

PONE-D-24-14484Physical and financial access challenges to seeking child healthcare in a rural district in GhanaPLOS ONE

Dear Dr. Nyande,

Thank you for submitting your manuscript to PLOS ONE. After careful consideration, we feel that it has merit but does not fully meet PLOS ONE’s publication criteria as it currently stands. Therefore, we invite you to submit a revised version of the manuscript that addresses the points raised during the review process. 

We look forward to receiving your revised manuscript.

Kind regards,

Abigail Kusi Amponsah, PhD.

Academic Editor

PLOS ONE

Additional Editor Comments (if provided):

Reviewers' comments:

Reviewer's Responses to Questions

**Comments to the Author**

1. Is the manuscript technically sound, and do the data support the conclusions?

Reviewer #1: Yes

Reviewer #2: Yes

2. Has the statistical analysis been performed appropriately and rigorously? 

Reviewer #1: N/A

Reviewer #2: Yes

3. Have the authors made all data underlying the findings in their manuscript fully available?

Reviewer #1: Yes

Reviewer #2: Yes

4. Is the manuscript presented in an intelligible fashion and written in standard English?

Reviewer #1: Yes

Reviewer #2: Yes

5. Review Comments to the Author

Reviewer #1: The manuscript is very well written.

-methodology part has clarity in terms of how the participants were recruited, how data recorded and analyzed. Particularly use of criterion of credibility, dependability, transferability, and confirmability is important that many manuscripts lack.

-analysis is presented in a manner that suitably answers the study objectives. only thing I would ask authors is to include a table listing codes that emerged during data analysis. Including such detail will further strengthen the manuscript as readers would be able to see how codes led to emergence of themes.

-discussion is quite detailed comparing findings with those reported in literature.

Reviewer #2: Physical and financial access challenges to seeking child healthcare in a rural district in Ghana: A review

INTRODUCTION

1. The words aimed to develop abbreviations should all be in capital letter e.g. LMICs in line 66 is sourced from lowercase letters. This also applies to similar cases within this article.

2. In literature review, most of the studies are locally by origin. Thus, they are mostly from Ghana. It will be better if some studies out of Ghana are added to this work to put this study in international context.

3. The physical and financial aspects can be summarized to be access. You may think of improving your objective and tittle to be more inclusive by mentioning of access to include both physical and financial aspects.

4. The presentation of core aspects of this study presents serious inconsistency in writing. For instance, in abstract the objective is “To explore the experiences of nurses and caregivers about the physical and financial 30 challenges to accessing child healthcare services in a rural district in Ghana” while in main article is “experiences of nurses and caregivers about accessibility challenges that impeded access to child healthcare services”. This is further written differently in line 81-82 presenting a slightly different phrase. In line 189, there is completely different thing but aiming on the same meaning. While they might be closely related, they are not the same. The core message should be consistent across the article.

METHODOLOGY

5. There are five qualitative designs including Ethnography, Narratives, Grounded theory, Phenomenology and Case study. In this study, explorative case descriptive design is used. There must be appropriate mentioning of among the well documented design.

6. In line 81, it is well understood that an interview is done to a single person and therefore there is no need of stating that it is individual interview.

7. Methodology section is poorly arranged. For instance, it is not normal to mention recruitment of study participants before even mentioning study population. It is common to first have study design followed by description of study are, study population, sampling and then followed by recruitment of participants. This is not the case in this work hence confounding flow and logic.

8. In my own understanding of your study, you have two study population which includes Nurses and caregivers. However, in your description from line 92-93, caregivers are not categorized as population but rather service users. In the following sentences, there is also evidence that caregivers are among study population.

9. Your sampling procedure is difficult to follow and thus can hardly be replicated elsewhere. In your sampling process, there is mentioning of levels of health facilities levels and numbers but there is no explanation on how these numbers were obtained from the facilities across study areas.

10. In line 107, there is mentioning of the status of individual which I think is not important in method section.

11. Data collection procedures do not show a place where interviews were carried out. In my own understanding a community is a group of families or households. In a situation like this where interviews are mentioned to have been done in community, it is difficult to know exact place of occurrence. One cannot tell if the interviews were conducted in household, church or at school compound.

12. In line 123 there is merely mentioning of saturation. You must explain the process in which saturation was attained. This will also need to show parallel processes of data collection and analysis.

13. Data management including quality control aspects are completely missing in the methodology section.

14. In my understanding, listening of the audio recording is part of data management in particular, quality control process. What was the necessity of doing so after fieldwork?

15. Data analysis approach is completely missing. There is no any approach such as thematic, content analysis and others forms, hence it is difficult to advise in absence of clear approach. However, there is mentioning of themes here but in a very simplified manner and cannot be replicated anywhere. This process must be well stipulated to enable the reader to understand the context in which the analysis was carried out.

16. The section of methodology needs a major revision including rearrangement and improvement of the contents in most subsections

RESULTS

17. This section presents themes and sub-themes which emerged from an unknown analysis approach which is not mentioned in analysis sub-section.

18. Line 189 presents a heading with new modified study attributes including health seeking behaviors which is not in your objective. You need to be consistent with terminologies used in this study.

19. A sub-them number 3.1 needs to be health facility barrier and not in category of physical barrier

20. In presentation of data, you must introduce the quote. In this work, the quotes emerge abruptly without invitation of a reader to see the quote

21. Most of the quotes are product of translation and therefore I did not expect to see words such as don’t, can’t etc, since they can only be extracted from direct quote and not through translation.

22. From line 205-210, there are three quotes illustrating one thing. I think one of the quotes could suffice the need. This also applies to many parts of your results where too many quotes representing a single scenario are common. Some quotes are also not thoroughly analysed.

23. Looking at your quotes and their equivalent analysis paragraphs, the quotes are taking too many spaces as compared to analysis. In most cases, there are more than two quotes presenting single aspect. Under this manner, there is danger that analysis is compromised since some quotes grouped together presents different understanding.

24. You must be consistent in writing particularly your headings. Some of your headings are written in a way that all words are capitalizes while other are presented in lowercase.

DISCUSSION

25. Your discussion is written in a monotonous way. Some paragraphs are prefaced by the word “again it emerges” and “it further emerges”. This must be rectified.

6. PLOS authors have the option to publish the peer review history of their article (what does this mean? ). If published, this will include your full peer review and any attached files.

**Do you want your identity to be public for this peer review?** For information about this choice, including consent withdrawal, please see our Privacy Policy .

Reviewer #1: **Yes: ** Dr. Imran Naeem

Reviewer #2: No

---

## [Author Response · Author response to Decision Letter 0]

18 Jan 2025

The name of the colleague who reviewed is Peter Agbezorlie

---

## [Decision Letter · Decision Letter 1]

11 Mar 2025

Physical and financial access challenges to seeking child healthcare in a rural district in Ghana

PONE-D-24-14484R1

Dear Dr. Nyande,

We’re pleased to inform you that your manuscript has been judged scientifically suitable for publication and will be formally accepted for publication once it meets all outstanding technical requirements.

Kind regards,

Abigail Kusi Amponsah, PhD.

Academic Editor

PLOS ONE

Additional Editor Comments (optional):

The two reviewers provided constructive feedback, and we appreciate your efforts in addressing their comments satisfactorily.

Reviewers' comments:

Reviewer's Responses to Questions

**Comments to the Author**

1. If the authors have adequately addressed your comments raised in a previous round of review and you feel that this manuscript is now acceptable for publication, you may indicate that here to bypass the “Comments to the Author” section, enter your conflict of interest statement in the “Confidential to Editor” section, and submit your "Accept" recommendation.

Reviewer #1: All comments have been addressed

Reviewer #2: All comments have been addressed

2. Is the manuscript technically sound, and do the data support the conclusions?

Reviewer #1: Yes

Reviewer #2: Yes

3. Has the statistical analysis been performed appropriately and rigorously? 

Reviewer #1: N/A

Reviewer #2: N/A

4. Have the authors made all data underlying the findings in their manuscript fully available?

Reviewer #1: Yes

Reviewer #2: Yes

5. Is the manuscript presented in an intelligible fashion and written in standard English?

Reviewer #1: Yes

Reviewer #2: Yes

6. Review Comments to the Author

Reviewer #1: (No Response)

Reviewer #2: The work appears to be well-structured now with a lot of improvement.

However there are following issues which must be addressed

1. The graphics in the table, which include codes and subthemes, should be improved to enable readers to clearly differentiate between codes aligned with specific sub-themes. To achieve this, each sub-theme should be presented in a separate table row, along with its respective codes, to clearly demarcate the codes for each sub-theme. Once this improvement is made, I suggest moving the entire Table 1 (which includes codes, subthemes, and themes) to the appendices, while keeping Table 2 (which contains only subthemes and themes) in its current position.

2.Regarding the analysis, each quote should be analyzed individually. The author should avoid grouping quotes together in one section and analyzing them as a whole. If the quotes carry a similar message, one should be selected as more illustrative, and the other should be omitted. If the quotes have distinct messages, each should be analyzed in separate paragraphs. These issues can be found in the following pages (lines 249-254, 274-280, 376-379, 420-424, 428-433, 440-446, 450-458, 467-473, 477-481, 488-495, 500-506, 539-542, and others). These need to be addressed appropriately.

3. The author should avoid using abbreviations that are not properly documented. For instance, in line 349, the term "hosp" is used, which is not standard English. This issue appears in several other instances as well.

4. There is a mismatch between the study design described in the abstract and the main document.

5. The author should select one of the five qualitative study designs (Case Study, Phenomenology, Narrative, Ethnography, or Grounded Theory). It is difficult to locate a study design referred to as "exploratory case descriptive design," which appears in the methods section. I believe the author intended to refer to a Case Study, but the wording is incorrect. Please make the necessary corrections to address this issue.

Once these issues are addressed the paper will be very good

7. PLOS authors have the option to publish the peer review history of their article (what does this mean? ). If published, this will include your full peer review and any attached files.

**Do you want your identity to be public for this peer review?** For information about this choice, including consent withdrawal, please see our Privacy Policy .

Reviewer #1: **Yes: ** Imran Naeem

Reviewer #2: No

---

## [Editor Report · Acceptance letter]

PONE-D-24-14484R1

PLOS ONE

Dear Dr. Nyande,

I'm pleased to inform you that your manuscript has been deemed suitable for publication in PLOS ONE. Congratulations! Your manuscript is now being handed over to our production team.

Kind regards,

on behalf of

Dr. Abigail Kusi Amponsah

Academic Editor

PLOS ONE